# Analysis of Electrochemical Performance with Dispersion Degree of CNTs in Electrode According to Ultrasonication Process and Slurry Viscosity for Lithium-Ion Battery

**DOI:** 10.3390/nano12234271

**Published:** 2022-12-01

**Authors:** Jaehong Choi, Chaewon Lee, Sungwoo Park, Tom James Embleton, Kyungmok Ko, Mina Jo, Kashif Saleem Saqib, Jeongsik Yun, Minki Jo, Yoonkook Son, Pilgun Oh

**Affiliations:** 1Department of Smart Green Technology Engineering, Pukyong National University, 45, Yongso-ro, Nam-gu, Busan 48547, Republic of Korea; 2Department of Nanotechnology Engineering, Pukyong National University, 45, Yongso-ro, Nam-gu, Busan 48547, Republic of Korea; 3Department of Electrical Engineering, Chosun University, 309, Pilmun-daero, Dong-gu, Gwangju 61452, Republic of Korea

**Keywords:** conductive additive, morphology, lithium-ion batteries, carbon nanotubes, slurry viscosity

## Abstract

Lithium-ion batteries (LIBs) continue to dominate the battery market with their efficient energy storage abilities and their ongoing development. However, at high charge/discharge C-rates their electrochemical performance decreases significantly. To improve the power density properties of LIBs, it is important to form a uniform electron transfer network in the cathode electrode via the addition of conductive additives. Carbon nanotubes (CNTs) with high crystallinity, high electrical conductivity, and high aspect ratio properties have gathered significant interest as cathode electrode conductive additives. However, due to the high aggregational properties of CNTs, it is difficult to form a uniform network for electron transfer within the electrode. In this study, to help fabricate electrodes with well-dispersed CNTs, various electrodes were prepared by controlling (i) the mixing order of the conductive material, binder, and active material, and (ii) the sonication process of the CNTs/NMP solution before the electrode slurry preparation. When the binder was mixed with a well sonicated CNTs/NMP solution, the CNTs uniformly adsorbed to the then added cathode material of LiNi_0.6_Co_0.2_Mn_0.2_O_2_ and were well-dispersed to form a flowing uniform network. This electrode fabrication process achieved > 98.74% capacity retention after 50 cycles at 5C via suppressed polarization at high current densities and a more reversible H1-M phase transition of the active material. Our study presents a novel design benchmark for the fabricating of electrodes applying well-dispersed CNTs, which can facilitate the application of LIBs in high current density applications.

## 1. Introduction

Lithium-ion batteries (LIBs) are continually increasing in demand due to their energy storage abilities, with high expectations set for their future performance [1,2]. LIBs have been applied in a variety of applications due to their success in mass production, from portable electronic devices to electric vehicles and energy storage systems [3]. LIBs have further succeeded in commercialization thanks to the continuous development of innovative materials [4] and electrode manufacturing technologies [5]. These materials and technologies affect the overall performance and characteristics of LIBs along with the price and market size which can be particularly valuable to the industry [4]. It is known that the performance of a battery is heavily dependent on the cathode active material [6,7,8], but the electrode is also crucial to maximizing the overall potential of these cathode active materials. A key performance limiting factor of the active material is its poor electrical conductivity, which is an issue applicable to layered cathode structures, as they have low intrinsic electrical conductivity [9,10]. Hence, this low electrical conductivity phenomena is witnessed in Ni-rich materials, with the electrical conductivity of the cathode material decreasing as the Ni content increases [11]. To combat this, conductive additives are included to increase the electrical network within the cathode composite.

Since the conductive material is not an electrochemically active material, it is preferable to apply it sparingly to maximize the electrode specific capacity. It is therefore better to have a high surface area material for the electron transfer in the electrode [12] Hence, carbon-based materials with relatively low atomic weight have been mainly used as conductive materials. Particularly, carbon black (CB), which is inexpensive, has excellent electrical conductivity and stable electrochemical properties [13]. CB is a 0 dimension material [14] and and is a form of amorphous carbon with a low aspect ratio and low electrical conductivity comparative to other carbons. These disadvantages make it unsuitable for batteries that require high power density, with the electron movement being a limiting factor at high C-rates. Therefore, research using carbon nanotubes (CNTs) with high aspect ratio and high crystallinity as a conductive material have been widely reported [13,14,15,16], but due to the high aggregation characteristics of CNTs during the electrode fabrication process, it is difficult to apply it effectively as a conductive material [17,18,19]. Therefore, focusing on finding an effective electrode fabrication process that combats this aggregation of CNTs will yield a significantly improved high current density performance for the cathode electrode.

Currently, the majority of LIB electrodes are produced using a wet slurry process that involves mixing the active material, binder, and conductive material in a liquid solvent often simultaneously, with the resulting mix then being coated on a current collector of metal foil. However, importantly, the composition [20] and order of addition [21,22,23] for the active materials, conductive materials, and binders strongly influences the electrode morphology [5], and variations of these can no doubt have an impact on the dispersion of the CNTs, resulting in improved capacity and cyclability of the batteries [24]. Research such as studies on the effect of the electrode drying time have proven the importance of the fabrication process on the carbon conductive network, with the lighter carbon additives rising to the electrode surface during the evaporation of the solvent and hence forming non-uniform carbon dispersions [25]. To combat this, the optimization of the slurry composition combined with the cell fabrication process is important to achieve a well-dispersed carbon network capable of suitable high current density performance [26].

In this work, we propose a strategy to suppress the aggregation of CNTs and to form a more uniform network of CNTs within the electrode, achieving improved electrical performance with long cycle life in high current density applications. After increasing the degree of dispersion of commercial CNTs/NMP solution through a facile ultrasonication treatment, a binder was mixed in to increase the viscosity of the CNTs solution and suppress the tendency of the CNTs to agglomerate during the electrode fabrication process. Four types of electrodes were examined via variation of the following conditions: (i) the mixing order of the conductive material, binder, and active material and (ii) the sonication of the CNTs solution pre slurry preparation. The process that most effectively achieved the dispersion of CNTs during electrode fabrication involved the binder being mixed first with the ultrasonicated CNTs solution. The CNTs were uniformly adsorbed to the then added cathode material of LiNi_0.6_Co_0.2_Mn_0.2_O_2_ and were dispersed to form a flowing uniform electrical conductivity network. By increasing the electrical conductivity of the electrode through this uniform dispersion of CNTs, the polarization was suppressed, even at increased current density. Additionally, the excellent cycle ability (>98.74% capacity retention after 50 cycles at 5C) was helped by the fact that the uniform network of CNTs, at 1.5 wt% in the electrode, stably maintained its average voltage during charge and discharge and a reversible H1-M phase transition of the active material was achieved. The morphology properties of CNTs on active materials were analyzed using scanning electron microscopy (SEM).

## 2. Experimental

### 2.1. Electrode Slurry Preparation Method

According to a molar ratio of 6:2:2 (Ni: Co: Mn transition metal ions), nickel sulfate (NiSO_4_·6H_2_O), cobalt sulfate (CoSO_4_·7H_2_O), and manganese sulfate (MnSO_4_·2H_2_O) were weighed, added into deionized water and stirred homogeneously to prepare a mixed solution of total metal ion concentration at 0.56 mol/L. The mixed solution was placed in a continuously stirring tank reactor (CSTR) for the coprecipitation reaction with 1 mol/L NaOH as the precipitator and 0.5 mol/L ammonia solution as the complexing buffer being added continuously. The resulting reaction process was completed in an N_2_ atmosphere to avoid air exposure. During the reaction, the pH value of the mixed solution was maintained between 10.5 to 11.5, and the temperature of reactor was controlled at 50 °C under continuous stirring for 24 h. To obtain Ni_0.8_Co_0.1_Mn_0.1_(OH)_2_ precursor, the coprecipitated particles were separated via filtration prior to washing and vacuum drying at 120 °C for 24 h. This precursor was then mixed with LiOH·H_2_O at a molar ratio of 1:1.03 and preheated at 500 °C for 5 h prior to calcination at 800 °C for 15 h, resulting in the Ni-rich layered oxide LiNi_0.6_Co_0.2_Mn_0.2_O_2_ (NCM622).

Carbon nanotubes (CNTs) dissolved in NMP solution were obtained from Cnano (LB108-43 NMP base conductive paste). Polyvinylidene fluoride (PVDF) was obtained from Sigma Aldrich and mixed in NMP at 8 wt% (*w*/*w*) for 20–24 h stirring at 200 rpm until the solid content was dissolved in the NMP. In certain procedures, a preliminary sonication process was applied to the CNT and NMP solution, before the electrode fabrication. In certain case the CNTs/NMP solution was diluted with NMP to achieve the corresponding solid contents wt% before sonication. In both cases, the sonication was conducted for 15 min. The corresponding electrode manufacturing process was then conducted in two different ways. Process 1: The CNT/NMP solution (1.5 wt% of the total electrode mass) was homogenized in a rotary mixer at 1500 rpm for 10 min before the addition of the active material (96.5 wt%) with a further 10 min of rotary mixing at 1500 rpm. Lastly, the 8 wt% in solution binder was added (to achieve 2 wt% of PVDF in electrode) followed by rotary mixing for 10 min at 1500 rpm. Process 2: The secondary process begins in a similar way with the CNT/NMP mix (1.5 wt%) being homogenized in a rotary mixer at 1500 rpm for 10 min. In this procedure, the addition of the 8 wt% in solution binder material (to achieve 2 wt% of PVDF in electrode) was followed by 10 min of rotary mixing at 1500 rpm. Lastly, the active material was added (96.5 wt%) with the final stage of rotary mixing being conducted at 1500 rpm for 10 min.

### 2.2. Electrode Fabrication Method

The cathode electrode composite prepared by the corresponding process comprised active material (NCM622), poly (vinylidene fluoride) (PVDF) binder, and the coated CNT as the conductive material in a ratio of 96.5:2:1.5. The eventual cathode electrode was then finalized via slurry casting onto aluminum foil with a manual doctor blade to active material loading at ~7 mg cm^−2^. This electrode was then dried for 50 min at 150 °C followed by pressing at 70 MPa, with a final drying process being conducted under vacuum at 120 °C for 6 h. The electrodes were then punched into 15 mm diameter size, in preparation for application in coin cells.

### 2.3. Coin Cell Fabrication Method

The electrolyte used was 1.3M LiPF_6_ in EC/EMC/DEC (=3/5/2,*v*/*v*/*v*) + 10% FEC + 0.5% VC + 1% PS + 0.2% LiBF_4_. The electrochemical data were attained via half-cell coin cell testing (2032R). The separator use was tri-layer PP/PE/PP (Celgard 2325). The counter electrode applied in the half cell was Li-metal at 0.5T.

### 2.4. Electrochemical Characterization Method

Standard Conditions: The formation cycle (first charge/discharge) was conducted at 0.1C and the voltage range was set at 2.75–4.3 V. The C rate was then increased to 5C for the cycle data, with the voltage range remaining consistent at 2.75–4.3 V. The temperature of all formation and cycling tests was 30 °C.

Rate Conditions: The conditions applied for the formation cycle were consistent at 0.1C and the voltage range was set at 2.75–4.3 V. For the rate test cycling the voltage range remained the same, with the charge rate being increased and fixed at 0.2C throughout all the cycles. However, the discharge rate was maintained at 0.2C for the first 3 cycles, followed by 0.5C, 1C, 2C, 3C and 5C for 3 cycles each, totaling 18 cycles with a steady increase in discharge rate. The temperature of all formation and cycling tests was 30 °C.

### 2.5. Microscopy Characterization Method

Scanning electron microscopy, SEM TESCAN (VEGA II LSU), was used to identify the morphology of CNTs’ coverage on the surface of the active material. Back-scattered electron (BSE) mode was applied to highlight the contrast in color between the NCM622 and CNTs, due to the electron dense active material. The SEM data ere collected for the entire cast electrode as opposed to powder formed samples, so that precise electronic pathways between particles could be observed as they would be in cell.

## 3. Result&Discussion

CNTs are initially well dispersed in the NMP solution, but when they are first mixed with cathode materials during the electrode fabrication process, the aggregation of CNTs begins to occur. In process1, the binder is added to the slurry of CNTs and cathode materials. However, when it is first mixed with the CNTs/NMP solution by changing the slurry process sequence, the viscosity in the slurry is increased, which restricts the movement of CNTs and suppresses their aggregation [27]. When the cathode materials were mixed into the binder/CNTs slurry, the aggregation of the CNTs is suppressed and it then mixed well with the cathode materials. This successful methodology will henceforth be denoted as process2. Even if CNTs are well dispersed in the solvent, aggregation still occurs in the solvent to some extent over time. Therefore, it is expected that aggregation of well-dispersed CNTs will be suppressed when mixed with the binder after the ultrasonic processing, achieving a high dispersibility of CNTs in solvent. Hence, four types of slurry were prepared by varying the two conditions in the electrode manufacturing process: (i) the mixing order of the binder, CNTs/NMP solution and active material during the fabrication of the electrode; and (ii) the sonication of the CNTs/NMP solution before fabrication of the electrode (Figure 1). Thus, a combination of these two factors resulted in four types of electrode: without sonication and process1 (w/o S&P1), with sonication and process1 (w/S&P1), without sonication and process2 (w/o S&P2), and with sonication and process2 (w/S&P2).

Electrodes were manufactured individually with the four slurries. The morphology and coverages of CNTs adsorbed to the cathode materials in the electrodes were analyzed using scanning electron microscopy (SEM). Figure 2 shows the SEM images in BSE mode, presenting the top view of the electrodes prepared via the four methods with 60 wt% of solid content in the slurry. The SEM images in BSE mode show higher atomic number materials as brighter, while the lower atomic number materials appear as darker [28,29]. Hence, the transition metal oxide cathode materials are bright and CNTs are relatively dark in the SEM images. Figure 2a shows w/o S&P1, in which CNTs are strongly agglomerated. Figure 2b and c are samples w/S&P1 and w/o S&P2, respectively. Although some of CNTs are agglomerated, the size of the CNT bundle is relatively reduced compared to w/o S&P1, resulting in increasing coverage of CNTs adsorbed to the active material. Lastly, in the case of w/S&P2 (Figure 2d), the degree of CNTs bundles on the electrode surface was greatly reduced, and the coverage was relatively increased, as shown by the darkening of the active material in the SEM imaging. This trend was emphasized when observing the zoomed-in SEM images (Figure 2e–h).

The electrochemical properties of the four electrodes were evaluated in LIB systems applying cathode materials of NCM622 to compare the initial discharge capacity and cycle properties. In Figure 3a, the initial formation (first charge/discharge) electrochemical evaluation at 0.1C within the potential range at 2.75–4.3 (V vs. Li/Li^+^) for electrodes of w/o S&P1, w/S&P1, w/o S&P2, and w/S&P2 show charge capacities of 198.8, 196.8, 197.2, and 197.3 mAh/g, and discharge capacities of 184.0, 182.0, 181.4, and 182.0 mAh/g, respectively. The initial coulombic efficiency (ICE) are 92.6, 92.5, 92.0, and 92.2%, respectively (Table 1). Formation data for a cell without carbon additive have been included in the Supplementary data (Appendix A). In the initial charge/discharge electrochemical evaluation, the charging capacity, discharging capacity, and ICE of the four electrodes were similar, which indicates that the electrode fabrication process did not significantly affect performance at the low current density. However, in the cycle data conducted at a high current density of 5C (Appendix A), there was a significant difference in performance depending on whether the ultrasonication was attempted and depending on the selected process sequence. Figure 3b shows the results of the electrochemical evaluation of the four electrodes over 50 cycles at a current density of 5C. In the case of w/o S&P1 electrodes, the discharge capacity of the first cycle was 142.2 mAh/g and the retention rate was 90.1% after 50 cycles, which indicated the lowest electrochemical performance of four electrodes. Interestingly, the initial capacity of the first cycle at 5C of the w/o S&P2 was 154.5 mAh/g, which is attributed to suppressing the aggregation of CNTs during the electrode fabrication by mixing the binder first with the CNTs/NMP solution. However, the retention of the w/o S&P2 cell after 50 cycles was 90.1%, which was the same as w/o S&P1 sample. When the w/S&P1 electrode is applied, which was mixed with binder immediately after the ultrasonication process, the discharge capacity of the first cycle is increased compared to w/o S&P1, and the retention rate after 50 cycles was also increased. In the case of the w/S&P2 sample, the discharge capacity of the first cycle was 159.1 mAh/g and the retention rate was 95.5%, resulting in excellent cycle performance. There was little difference in the coulombic efficiency among the electrodes during cycling at a current density of 5C (Figure 3c). The CNTs network in the electrode could be well formed by improving the dispersion degree of CNTs as achieved through the ultrasonication process. The CNTs aggregation was suppressed during electrode fabrication by mixing the CNTs/NCM solution with the binder to increase the viscosity of the slurry.

Figure 4a displays the rate capability test of w/o S&P1, w/S&P1, w/o S&P2, and w/S&P2 at various discharge current rates within the potential range of 2.75–4.3 (V vs. Li/Li^+^). During the rate performance the charge current density was isolated at 0.1C, with the discharge current varying. The discharge capacity of w/o S&P1, w/S&P1, w/o S&P2, and w/S&P2 at 5C is 155, 160, 157, and 169 mAh/g, respectively. In the case of w/o S&P1, the discharge capacity was higher than w/S&P1 and w/o S&P2 at 0.2C. As the current density increased, however, the discharge capacity gradually decreased. Hence, the lowest discharge capacity was exhibited at the current density of 5C. In the case of w/S&P2, the discharge capacity at 0.2C was slightly higher than that of w/o S&P1, but the decrease in the discharge capacity was low compared to the other samples with the increase of the current density. Hence, similar to the cycle performance trend, the w/S&P2 showed the greatest rate capability. Moreover, the charge-discharge curve of w/o S&P1, w/S&P1, w/o S&P2, and w/S&P2 at the different currents exhibited voltage polarization (Figure 4b–e), with additional comparisons of these data being included in Appendix A. The largest and the lowest polarization voltages were exhibited from w/o S&P1 and w/S&P2 displaying 0.022 V, 0.087 V, 0.125 V, 0.203 V, and 0.022 V, 0.072 V, 0.111 V, and 0.118 V at 0.2C, 2C, 3C and 5C, respectively. These voltages were calculated as the difference between the final charge voltage and the initial discharge voltage of the formation cycle, which equates to the voltage drop during the 10-min rest interval where no current is applied. The improved electrochemical performance in the high current density indicates that the internal resistance of the electrode is reduced, which means that the CNTs form an effective network for electron transfer in the electrode [14,30]. Hence, this indicates that the w/S&P2 has a lower internal electrode resistance than the w/o S&P1, and the CNTs network of w/S&P2 is well formed compared with w/o S&P1 [14,30].

To form a highly uniform CNTs network in the electrode, the electrode was manufactured by directly mixing the binder materials after maximizing the degree of dispersion of CNTs through the ultrasonic process. To confirm the effect of the ultrasonication process, the process was tested at differing solid contents (SC) conditions. Figure 5a shows the formation voltage profile at 0.1C of the w/o S&P1 and w/S&P2 electrode fabricated at 60 wt% of SC, where the results were obtained from the previously presented Figure 2 results, with further w/S&P2 electrodes fabricated at 40 wt% and 50 wt% of SC. In the formation electrochemical evaluation at 0.1C, the SC effect on the discharge capacity was not conclusive (Table 2). However, in the cycle performance at high current density (Figure 5b and Appendix A), the retention improved to more than 98% after 50 cycles with similar discharge capacity in the first cycle for the samples of 40% and 50% of SC. This indicates that the ultrasonication effect was more pronounced at low SC, resulting in a well-dispersed CNT electrode. This effect results from the lower viscosity of the low SC samples allowing the sonication process to disperse the CNTs more efficiently within the CNTs/NMP solution. This confirms that dispersion of CNTs in the CNTs/NMP solution is the key benefit of the ultrasonication process. Regardless of the SC, the coulombic efficiencies were similar for all four samples (Figure 5c). Lower coulombic efficiency is shown for the first cycle at 5C as a result of the large rate increase from the formation charge/discharge (0.1C) [31]. The average charge/discharge voltages of w/o S&P1 (SC: 60 wt%), w/S&P2 (SC: 60 wt%), w/S&P2 (SC: 50 wt%), and w/S&P2 (SC: 40 wt%) electrodes during cycle evaluation are shown in the Figure 5d. When the solid content was low, the ultrasonication-treated electrodes showed a reduced overvoltage. The relatively low charging and high discharging voltage, as shown best in the w/S&P2 (SC: 50 wt%) electrode, indicates that the network for electron transfer was well formed in this electrode.

To better understand the effect of the electron transfer network on the charge and discharge properties of w/o S&P1 (SC: 60 wt%), w/S&P2 (SC: 60 wt%), w/S&P2 (SC: 50 wt%), and w/S&P2 (SC: 40 wt%), the differential capacity (dQ/dV) curves were analyzed using the cycle evaluation data at current density of 5C (Figure 6a–d). The dQ/dV curves usually display a triplet of redox peaks due to the phase transitions from hexagonal (H1) to monoclinic (M), monoclinic (M) to hexagonal (H2), and hexagonal (H2) to hexagonal (H3) during the delithiation and lithiation processes within potential range of 2.75–4.3 (V vs. Li/Li^+^) [32]. However, due to the high current density, the H2 peak and H2-H3 phase transition did not appear in the dQ/dV curve. As numbers of cycle increased, the H1-M oxidation peak shifted towards higher potentials in all of the electrodes, with such movement being expected, as the NCM622 was bare and unprotected. In the case of w/o S&P1 (SC: 60 wt%), the shift of the H1-M oxidation peak is large, and the shift of the H1-M oxidation peak is low in the w/S&P2 (SC: 50 wt%) electrode. This result indicates that w/S&P2 (SC: 50 wt%) electrode exhibited a more stable reversibility for the phase transition of H1-M during cycling than that of others which can be attributed to the conductivity network maintaining contact with the active material throughout cycling.

## 4. Conclusions

In conclusion, we fabricated well-dispersed CNTs coated cathode electrodes for high current density applications with excellent cycle performance and rate capability. The CNTs in the CNTs/NMP solution were dispersed during sonication treatment and prevented from forming CNTs bundles during fabrication of the electrode through direct mixing with binder materials to increase the viscosity of slurry. The degree of dispersion of CNTs in the electrode was an indicator of the formed electron transfer network which was directly related to the overall electrochemical performance of the LIBs. When the solid content is controlled at 50 wt%, the ultrasonication effect is more pronounced with improved electrochemical performance. Hence, in the electrochemical evaluation at a current density of 5C, the discharge capacity of the first cycle is 158.9 mAh/g and the retention at the fiftieth cycle is 98.74%, achieving excellent electrochemical performance using CNTs as conductive materials at 1.5 wt%. When the CNTs were well dispersed and formed a uniform electron transfer network in the electrode, polarization was suppressed at high current densities, and an excellent electrochemical performance was achieved due to the reduced average charge voltage and increased average discharge voltage during cycle evaluation. In addition, the improved electron transfer network helped the stable reversibility of the H1-M phase transition of the cathode material at high current densities. We hope our study helps with the fabrication of cathode electrodes with well-dispersed CNTs for high power current density applications using LIBs systems.

## Figures and Tables

**Figure 1 nanomaterials-12-04271-f001:**
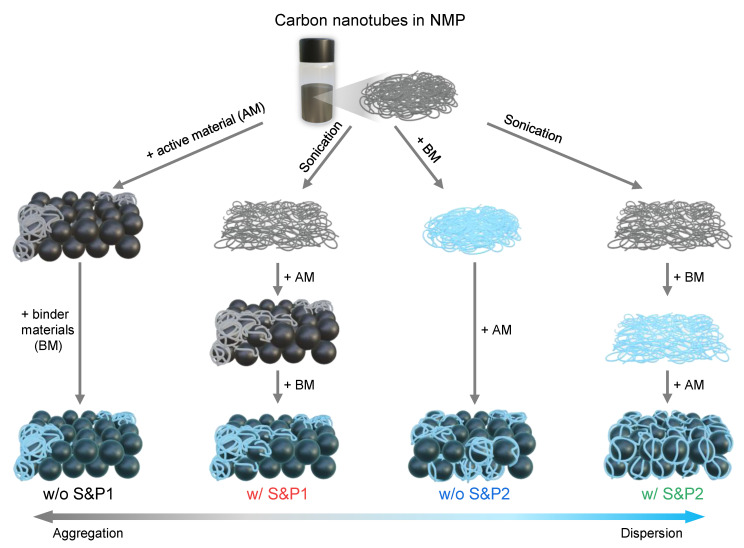
Scheme of preparation routes for the four types of electrode made by varying two conditions: (i) the mixing order of the conductive material, binder and active material and (ii) the sonication process of the CNTs/NMP solution before the electrode slurry preparation.

**Figure 2 nanomaterials-12-04271-f002:**
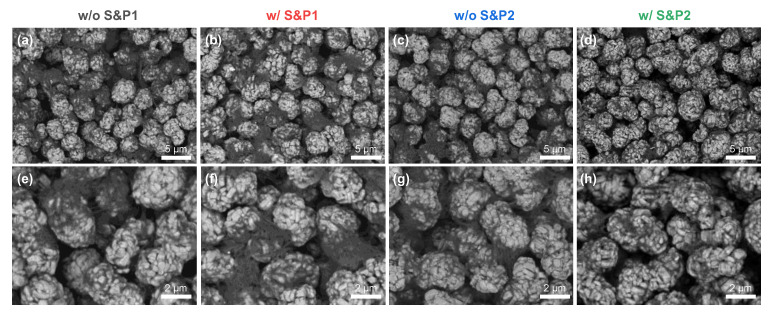
(**a**–**d**) Scanning electron microscopy (SEM) images of electrode surface and (**e**–**h**) zoomed-in image of without S&P1, with S&P1, without S&P2, and with S&P2, respectively.

**Figure 3 nanomaterials-12-04271-f003:**
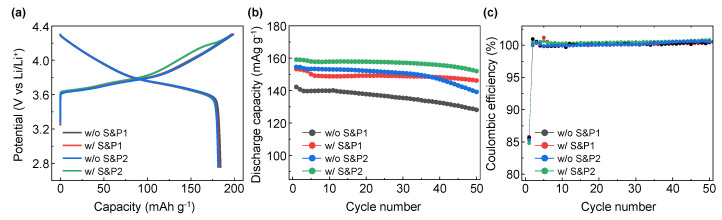
(**a**) Electrochemical performance of without S&P1, with S&P1, without S&P2, and with S&P2 in evaluation of lithium-ion battery (LIBs) for the formation (initial charge−discharge) voltage profiles at 0.1C rate between 2.75 and 4.3 V at 30 °C. (**b**) The following 50 cycle performance at 5C at 30 °C. (**c**) Coulombic efficiency of the 50 cycles with charge and discharge C-rates of 5C.

**Figure 4 nanomaterials-12-04271-f004:**
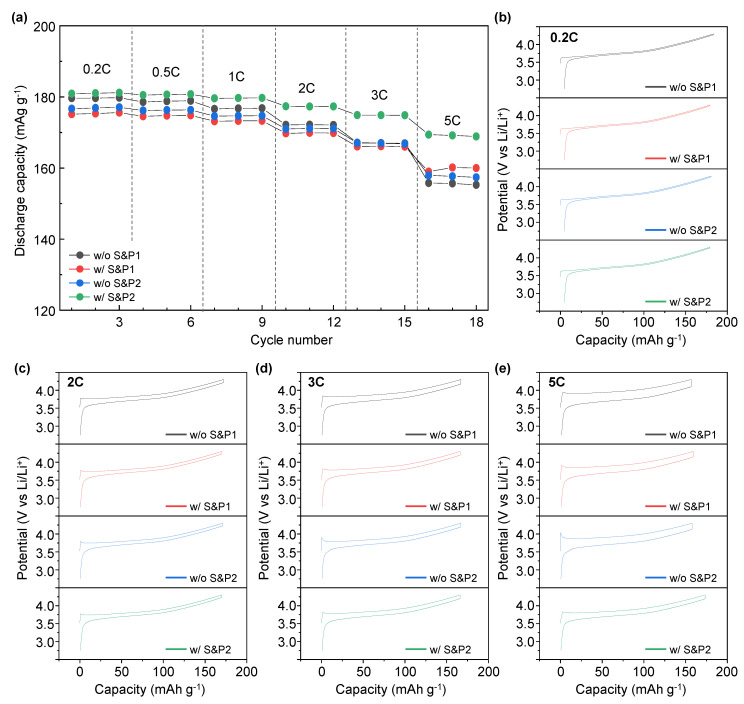
(**a**) Rate capability of without S&P1, with S&P1, without S&P2, and with S&P2 at cycle discharge current rates of 0.2C−5C within the potential range of 2.75–4.5 (V vs. Li/Li^+^). Voltage profile of (**b**) 0.2C, (**c**) 2C, (**d**) 3C, and (**e**) 5C showing charge and discharge cycles. The curves come from the rate data in (**a**).

**Figure 5 nanomaterials-12-04271-f005:**
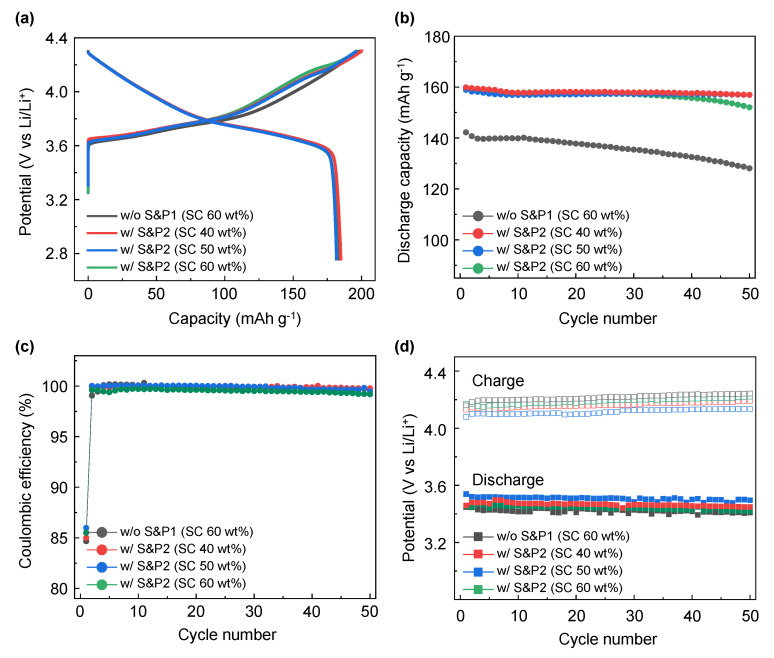
(**a**) Voltage profile of the with S&P2 electrode fabricated with different wt% of solid content (SC) for the formation (initial charge−discharge) voltage profiles at 0.1C rate between 2.75 and 4.3 V at 30 °C. The result of without S&P1 (SC 60 wt%) and with S&P2 (SC 60 wt%) are obtained from Figure 3. (**b**) the following 50-cycle performance at 5C at 30 °C. (**c**) Coulombic efficiency with charge and discharge C-rates of 5C. (**d**) Average voltage changes over the 50 cycle tests.

**Figure 6 nanomaterials-12-04271-f006:**
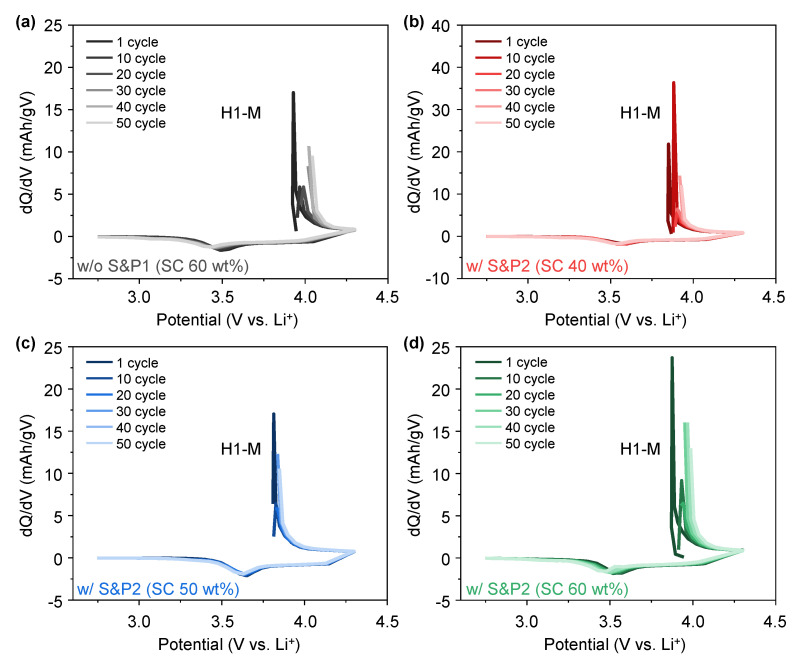
dQ/dV analysis for the cycle charge−discharge of cells containing (**a**) without S&P1 (SC 60 wt%), (**b**) with S&P2 (SC 40 wt%), (**c**) with S&P2 (SC 50 wt%), and (**d**) with S&P2 (SC 60 wt%) at 5C.

**Table 1 nanomaterials-12-04271-t001:** Formation charge capacity, discharge capacity, and initial coulombic efficiency (ICE) of without S&P1, with S&P1, without S&P2, and with S&P2 at 0.1C at 30 °C. Cycle charge and discharge capacity of 1 cycle, 50 cycle, and capacity retention after 50 cycle at 5C.

	Charge Capacity (mAh/g)	Discharge Capacity (mAh/g)	ICE	Discharge Capacity (mAh/g) @1clycle	Discharge Capacity (mAh/g) @50clycle	Discharge Retention (%) @50clycle
w/o S&P1	198.8	184.0	92.6	142.2	128.1	90.1
w/S&P1	196.8	182.0	92.5	153.3	146.2	95.4
w/o S&P2	197.2	181.4	92.0	154.5	139.2	90.1
w/S&P1	197.3	182.0	92.2	159.1	152.0	95.5

**Table 2 nanomaterials-12-04271-t002:** Charge capacity, discharge capacity and initial coulombic efficiency (ICE) of without S&P1 (SC 60 wt%), with S&P2 (SC 40 wt%), with S&P2 (SC 50 wt%), and with S&P2 (SC 60 wt%) at 0.1C at 30 °C. Charge and discharge capacity at 1 cycle, 50 cycle, and capacity retention after 50 cycles at 5C.

	Charge Capacity (mAh/g)	Discharge Capacity (mAh/g)	ICE	Discharge Capacity (mAh/g) @1clycle	Discharge Capacity (mAh/g) @50clycle	Discharge Retention (%) @50clycle
w/o S&P1	198.8	184.0	92.6	142.2	128.1	90.1
w/S&P2 (SC 40 wt%)	200.7	184.9	92.1	159.9	156.9	98.1
w/S&P2 (SC 50 wt%)	196.5	181.8	92.5	158.9	156.9	98.7
w/S&P1 (SC 60 wt%)	197.3	182.0	92.2	159.1	152.0	95.5

## Data Availability

Not applicable.

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
