# Peer review of "Analysis of Electrochemical Performance with Dispersion Degree of CNTs in Electrode According to Ultrasonication Process and Slurry Viscosity for Lithium-Ion Battery"

_nanomaterials, 2022, doi:10.3390/nano12234271_

Round 1

Reviewer 1 Report

1. The "Abstract" needs to be more concise.

2. The role of “Introduction” of the whole paper is to introduce the background of your research and attract more attention. However, this did not realize in this manuscript. The logic of Introduction is kind of messy, please re-write this part. Especially, the description from Line 98 to Line 109 is too much overlapped with the "Conclusions", which should not happen. 

3.  The difference under the current densities are not compared in Table 1.

4. From Line 271-Line 273, it is suggested that the authors should write the calculation formula directly, which cannot be obatined from the figure.

5. From Line 276-Line 278, how to calculate the internal electrode resistance, more details are needed here.

6. For Figure 4, if the authors could combine the sub-figures together, the readers could get the differences between them more straight.

7. Line 290, why the effect is almost negligible? I am confused.

8. I would like to suggest going through the whole manuscript, including the captions of all the figures and tables more carefully for clarity, syntax and correctness. The English should be improved for the sake of clarity.

Author Response

We are truly thankful for reviewers’ insightful comments, which enabled us to significantly improve the quality of our manuscript. In the following pages, we provide point-by-point responses to each of the reviewer comments. We have revised the manuscript in accordance with reviewer’s comments and suggestions and the request of the editorial office. Revisions in the manuscript are explicitly shown with yellow highlights in addition. 

Reviewer 2 Report

This paper is on improving the quality of lithium battery anode materials. Carbon nanotube was used as the additive to increase the charge speed of the carbon added electrodes. The advantages of using carbon nanotube as the current collector was shown.

The data support conclusion. It is recommended for publishing.

Author Response

We are truly thankful for reviewers’ comments.

Reviewer 3 Report

The manuscript discuss, systematic study of homogeneous dispersion of electrically conductive CNTs over LiNi0.6Co0.2Mn0.2O2 (NMC) cathode in Lithium-ion batteries, and also studied the optimization of solid content in the cathode slurry. The optimized NMC cathode showed an initial discharge capacity of 158 mAh g-1 with 98.74% capacity retention after 50 cycles at a 5C rate. The author claims that the uniform dispersion of CNTs increases the electrical conductivity of the electrode, and suppresses the polarization even at higher current densities. However, the results are interesting, it is put forward to address the below quarries to author,

1. In SEM images CNT distribution of w/S&P2 are not clear, elemental mapping might give a clear difference.

2. For better comparison in Figure 3b, discharge capacity of cells without CNTS are missing

3. In figures 3 &5 the difference in voltage profile is not obvious at a lower current (0.1C), and also total figure discuss at 5C current densities, it is suggested to include voltage profile at higher current densities (5C)

4. Reason behind low coulombic efficiency in the initial cycles and why different coulombic efficiencies are observed with different wt% of solid content (Figure 5c) with the same cathode can be included in the manuscript.

5. Below references might be consider to the manuscript.

a) Appl. Surf. Sci. 2021,568, 150934
b) Chem electro chem, 2020, 7, 12, 2621-2628
c) ACS Appl. Energy Mater. 2020, 3, 11, 10872–10881

Author Response

(The authors gave the same response as above.)

Round 2

Reviewer 1 Report

Accept

Reviewer 3 Report

The author addressed all the questions. Now, manuscript can be accepted in the present form.